# Cooperative Heterogeneous Deep Reinforcement Learning

**Han Zheng**
AAII,University of Technology Sydney
Han.Zheng-1@student.uts.edu.au

**Pengfei Wei**
National University of Singapore
wpf89928@gmail.com

**Jing Jiang**
AAII,University of Technology Sydney
jing.jiang@uts.edu.au

**Guodong Long**
AAII,University of Technology Sydney
guodong.long@uts.edu.au

**Qinghua Lu**
Data61, CSIRO
qinghua.lu@data61.csiro.au

**Chengqi Zhang**
AAII,University of Technology Sydney
Chengqi.Zhang@uts.edu.au

## Abstract

Numerous deep reinforcement learning agents have been proposed, and each of them has its strengths and flaws. In this work, we present a Cooperative Heterogeneous Deep Reinforcement Learning (CHDRL) framework that can learn a policy by integrating the advantages of heterogeneous agents. Specifically, we propose a cooperative learning framework that classifies heterogeneous agents into two classes: global agents and local agents. Global agents are off-policy agents that can utilize experiences from the other agents. Local agents are either on-policy agents or population-based evolutionary algorithms (EAs) agents that can explore the local area effectively. We employ global agents, which are sample-efficient, to guide the learning of local agents so that local agents can benefit from sample-efficient agents and simultaneously maintain their advantages, e.g., stability. Global agents also benefit from effective local searches. Experimental studies on a range of continuous control tasks from the Mujoco benchmark show that CHDRL achieves better performance compared with state-of-the-art baselines.

## 1   Introduction

Deep reinforcement learning (DRL) integrates deep neural networks with reinforcement learning principles, e.g.,Q-learning and policy-gradient, to create a more efficient agent. Recent studies have shown a great success of DRL in numerous challenging real-world problems, e.g., video games and robotic control [19]. Although promising, existing DRL algorithms still suffer from several challenges including sample complexity, instability, and temporal credit assignment problems [28, 9].

One popular research line of DRL is policy-gradient based on-policy methods attempting to evaluate or improve the same policy that is used to make decisions [29], e.g., trust region policy optimization (TRPO) [24] and proximal policy optimization (PPO) [25]. Recent works [31, 17] have proved that policy-gradient based methods can converge to a stationary point under some conditions, which theoretically guarantees their stability. However, they are extremely sample-expensive since they require new samples to be collected in each gradient step [29].

On the contrary, Q-learning based off-policy methods, which is another research line evaluating or improving a policy different from the one that is used to generate the behavior, can improve

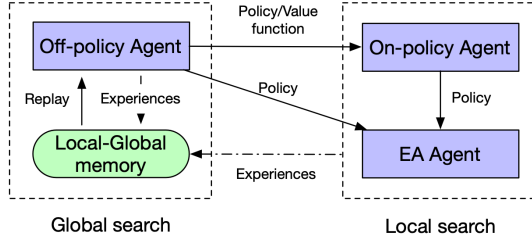

Figure 1: The high-level structure of CHDRL for one iteration

sample efficiency by reusing past experiences [29]. Existing off-policy based methods include deep Q-learning network (DQN) [19] and Soft Actor-Critic (SAC) [8] etc. These methods involve the approximation of some high-dimensional and nonlinear functions, usually through deep neural networks, which poses a significant challenge on convergence and stability [3, 9]. It is also well known that off-policy Q learning is not to converge even with linear function approximation [2]. Moreover, recent studies [14, 6] identify some other key sources of instability for off-policy methods, i.e., bootstrapping and extrapolation errors. As shown in [14], off-policy methods are highly sensitive to data distribution, and can only make limited progress without exploiting additional on-policy data.

In addition to the pros and cons discussed above, on-policy and off-policy methods based on temporal difference learning suffer from some common issues. The one that received much research attention is the so-called *temporal credit assignment* problem [28]. When rewards become sparse or delayed, which is quite common in real-world problems, DRL algorithms may yield an inferior performance as reward sparsity downgrades the learning efficiency and hinders exploration. To alleviate this issue, evolutionary algorithms (EAs) [5, 26] have recently been introduced to DRL [21, 13]. The usage of a fitness metric that consolidates returns across the entire episode makes EAs indifferent to reward sparsity and robust to long time horizons [22]. However, EAs suffer from high sample complexity and struggle to solve high-dimension problems involving massive parameters.

In this paper, we are interested in an algorithm that takes the essence and discards the dross of different DRL algorithms to achieve high sample efficiency and maintain good stability in various continuous control tasks. To do so, we propose a framework called CHDRL. Specifically, CHDRL works on an agent pool containing three classes of agents: an off-policy agent, an on-policy agent, and a opulation-based EAs agent. All the agents cooperate based on the following three mechanisms.

Firstly, all agents collaboratively explore the solution space following a hierarchical policy transfer rule. As the off-policy agent is sample-efficient, we take it as the global agent to obtain a relatively good policy or value function at the beginning. The on-policy agent and the population-based EAs agent are taken as local agents and start their exploration with the prior knowledge transferred from the global agent. As the EAs agent is population-based, we further allow it to accept policies from the on-policy agent.

Secondly, we employ a local-global memory replay to enable global (off-policy) agents to replay the newly generated experiences by local (on-policy) agents more frequently so that global agents can benefit from local search. Note that, with policy transfer as stated above, local agents start exploration with a policy transferred from global agents, and thus their generated experiences can be taken as close to the on-policy data of global agents' current policy [14, 6]. By allowing global agents exploits more often from these local experiences, we can alleviate the bootstrapping or extrapolation error and further boost global agents' learning. Consequently, global agents provide a better starting point for local agents who in turn generate more diverse local experiences for global agents' replay, which forms a good win-win cycle.

Thirdly, although we encourage the cooperation among agents in exploration, we also tend to maintain the independence of each agent; that is, we do not want the learning of local agents to be completely dominated by that of global agents. This is to enable each agent to still maintain its policy updating scheme and preserve its learning advantage. To do so, we firstly develop a loosely coupled hierarchical framework with global agents at the upper-level and local agents at the lower-level[1]. Such a framework not only makes each agent generally run in a relatively independent environment

with different random settings, but also achieves the easy and flexible deployment or replacement of the agent candidates used in the framework. Secondly, to avoid over-policy-transfer, i.e., policy transfer happening too frequently thus interrupting the learning stability of local agents, we set a threshold to control the frequency of policy transfer.

The high-level structure of CHDRL is shown in Figure 1. In this work, we instantiated a CHDRL with PPO, SAC, and Cross-Entropy-Method (CEM) based EA [27], named CPSC. Experimental studies showed the superiority of CPSC to several state-of-the-art baselines in a range of continuous control benchmarks. We also conducted ablation studies to verify the three mechanisms.

## 2 Preliminaries

In this section, we review the representation of on-policy method, off-policy method, and EAs, namely, PPO [25], SAC [8], and Cross-Entropy based EA [27].

### 2.1 Proximal Policy Optimization (PPO)

PPO is an on-policy algorithm that trains a stochastic policy. It explores by sampling actions according to the latest version of its stochastic policy. During training, the policy typically becomes progressively less random, as the update rule encourages it to exploit rewards that it has already found. PPO tries to keep new policies close to old.

### 2.2 Soft Actor-critic (SAC)

SAC is an off-policy algorithm that incorporates an entropy measure of the policy into the reward to encourage exploration. The idea is to learn a policy that acts as randomly as possible while still being able to succeed in the task. It is an off-policy actor-critic model that follows the maximum entropy RL framework. The policy is trained with the objective of maximizing the expected return and entropy at the same time.

### 2.3 Evolutionary Algorithms and CEM-ES

EAs [5, 26] are a class of black-box search algorithms that apply heuristic search procedures inspired by natural evolution. Among EAs, Estimation of Distribution Algorithms (EDAs) are a specific family where the population is represented as a distribution using a covariance matrix [15]. CEM is a simple EDA where the number of elite individuals is fixed at a certain value. After all individuals of a population are evaluated, the top fittest individuals are used to compute the new mean and variance of the population.

## 3 Related Works

Experience replay mechanism [16] is widely used in off-policy reinforcement learning to improve sample efficiency. DQN [19] randomly and uniformly samples experience from a replay memory. [23] subsequently expands DQN to develop a prioritized experience replay (PER), which uses a temporal difference error to prioritize experiences. Zhizheng Zhang et al. [33] introduce an episodic control experience replay method to quickly latch on to good trajectories. Our local-global memory uses a different strategy: let the off-policy agent learn more from effective on-policy experiences.

CHDRL's cooperative learning mechanism can be discussed in terms of guided policy search (GPS) [7, 11] or evolutionary reinforcement learning (ERL) [21, 13, 12]. For GPS, they generally need to use the KL divergence to guide how policies are improved. ERL [13] directly transfers the RL agent's policy to the EA population, while Pourchot et al. [21] uses the RL's critic to update half of the EA population using the gradient-based technique. The proposed CHDRL is related to GPS and ERL in the sense that multiple polices work in a hybrid way. However, the main difference between CHDRL and other similar methods is how heterogeneous agents cooperate. Moreover, CHDRL can benefit not just from off-policy and EA learning schemes but also from the on-policy learning scheme.

Another related area of work is in the training architectures. A3C [18] introduce an asynchronous training framework for deep reinforcement learning, showing parallel actor-learners have a stabilizing

effect on training. Babaeizadeh et al. [1] adapt this approach to make efficient use of GPUs. IMPALA [4] uses a central learner to run SGD while asynchronously pulling sample batches from many actor processes. Horgan et al. [10] proposes a distributed architecture for training DRL that employs many actors to explore using different policies and prioritizing the generated experiences. Han Zheng et al. [34] introduces a training method to select the best agent for different tasks. All these methods only focus on one learning scheme, and/or all actors involved are treated equally. On the contrary, CHDRL distinguishes actors as global actors and local actors that serve for different purposes respectively. Moreover, CHDRL focuses on the cooperation of diverse learning schemes.

## 4 Cooperative Heterogeneous Deep Reinforcement Learning(CHDRL)

In this section, we firstly introduce the proposed CHDRL framework and then suggest a practical algorithm based on it. Our CHDRL mainly follows three mechanisms to achieve cooperative learning of heterogeneous agents: cooperative exploration (CE), local-global memory relay (LGM) and distinctive update (DU).

**Cooperative Exploration (CE)**. The key idea of CE is to utilize a sample-efficient agent, such as an off-policy agent, to guide the exploration of the agent with a relatively lower sample efficiency, e.g., an on-policy agent. This is done by transferring policies across agents. More precisely, the sample-efficient agent acts as a global agent and conducts a global search first. In every iteration, we want to use the policy and/or value function obtained by the global agent as the prior knowledge to re-initialize local agents so that they can start to exploit from a relatively better position. To do so, we need to solve three key points: what to transfer, how to transfer, and when to transfer, following the basic mechanism of transfer learning[20, 32].

*What to Transfer.* Different agents may have different policy architectures. The policy could be deterministic, where it is denoted by $a \doteq \mu_\phi(s)$, or stochastic, where it is denoted by $a \sim \pi_\phi(\cdot|s)$. In continuous control tasks, the stochastic policy is usually assumed to be sampled from a Gaussian distribution, and thus it can be represented as:

$$a \doteq \mu_\phi(s) + \Sigma$$

where $\mu_\phi(s)$ is the mean action, $\Sigma$ represent a covariance matrix. Typically, $\Sigma$ may have different forms, e.g., PPO uses a state-independent $\Sigma$ while SAC utilizes a state-dependent one. However, a similar mean policy architecture $\mu_\phi(s)$ is used in different methods. Inspired by this, we propose to use the structurally identical mean function $\mu_\phi(s)$ to establish a link between the deterministic and stochastic policies. Then the policy among heterogeneous agents can be shared by transferring $\mu_\phi(s)$.

*How to Transfer.* As shown in Figure 1, policies are transferred following a hierarchical manner. The principle is that policies are transferred from upper-level agents with higher sample efficiency to the lower-level agents with lower sample efficiency. More specifically, policies are transferred (1) from off-policy agents to both on-policy agents and EAs agents, and (2) from on-policy agents to EAs agents. Note that EAs agents are population-based, and thus we allow them to accept the on-policy agent's policy to maximize the transfer capacity. To avoid collisions, we use different individuals of EAs' population to accept policies from different upper-level agents. As EAs agents accept policies from both off-policy and on-policy agents, they naturally serve as a pool that stores all the transferred policies.

*When to Transfer.* Policy transfer happens only when upper-level agents find a better policy than the current one of lower-level agents. Lower-level agents then re-initialize the exploration with the policy transferred from their upper-level agents as the new starting point. In order to compare the performance of policies, we use the average return as the evaluation metric. To be statistically stable, we use the average return over five episodes as the policy's performance score. Moreover, to avoid that policy transfer happens too frequently to interrupt the learning stability of lower-level agents, we enable policy transfer only when the performance gap is larger than a predefined threshold.

**Local-Global Memory Relay (LGM)**: Off-policy agents can make more progress when considering on-policy data in their learning [14, 6]. Following this observation, we employ a local-global memory replay mechanism to enable global off-policy agents to benefit from diverse local experiences from both on-policy agents and EAs agents. In particular, we propose two memory buffers – a global one and a local one – to store the generated exploration experiences. The global memory serves to store the entire exploration experiences of all the agents, while the local memory only stores the

**Algorithm 1** CSPC

**Require:**

    $G_s$ with policy $\pi_s \doteq \mu_{\phi_s}(s) + \Sigma_s$ and value $\psi_s$; $L_p$ with $\pi_p \doteq \mu_{\phi_p}(s) + \Sigma_p$ and value $\psi_p$; local memory $M_l$, global memory $M_g$; Iteration steps $T$; $L_c$ with policies as $\mu_{\phi_{c_0}}(s), ..., \mu_{\phi_{c_n}}(s)$; initial steps $T_g$; gap $f$, terminate step $T_m$, and initial test score $S_s, S_p, S_c$. Initialize transfer label $A_p, A_c$ to False.

1: **repeat**
2:     **TRAIN**$(G_s, T_g)$, $t \leftarrow t + T_g$
3:     **for** Agent $a$ in $G_s, L_p, L_c$ **do**
4:         **TRAIN**$(a, M_l, M_g, T)$
5:         **if** $a$ is not $G_s$ **then**
6:             **UPDATE**$(\phi_s, M_l, M_g, T)$
7:         **end if**
8:         $t \leftarrow t + T$
9:     **end for**
10:    Update test scores $S_s, S_p$ and $S_c$
11:    **if** $S_s - S_p > f$ **then**
12:       $\phi_p \leftarrow \phi_s, \psi_p \leftarrow \psi_s, A_p \leftarrow$True
13:    **end if**
14:    **if** $S_s - S_c > f$ **then**
15:       $\phi_{c0} \leftarrow \phi_s, A_c \leftarrow$True
16:    **end if**
17:    **if** $S_p - S_c > f$ **then**
18:       $\phi_{c1} \leftarrow \phi_p$
19:    **end if**
20: **until** $t > T_m$

**Algorithm 2** TRAIN

**Require:**

    Input agent $a$, training steps $T_a$, episode reward $R = 0$, $R_m \leftarrow \min(S_s, S_p, S_c)$, step $t = 0, t_e = 0$, global memory $M_g$, local memory $M_l$ episode memory $M_e$.

1: **repeat**
2:     Observe state $s$ and select action $a \sim \mu_{\phi_s}(s) + \Sigma_s$ or $a \sim \mu_{\phi_p}(s) + \Sigma_p$ or $a \doteq \mu_{\phi_{c_i}}(s)$
3:     Execute $a$ in the environment
4:     Observe next state $s'$, reward $r$, and done signal $d$
5:     Store $(s, a, r, s', d)$ in $M_e$, $R \leftarrow r + R$
6:     $t \leftarrow t + 1, t_e \leftarrow t_e + 1$
7:     **if** $s'$ is terminal **then**
8:         $\phi' \leftarrow$ **UPDATE**$(\phi, M_g, M_l, t_e)$ where $\phi \in \{\phi_s, \phi_p, \phi_c\}$
9:         **if** $R > R_m$ and ($a$ is $G_s$ or $A_p$ or $A_c$ is True) **then**
10:        Store $M_e$ in $M_l$ and $M_g$
11:       **else**
12:        Store $M_e$ in $M_g$
13:       **end if**
14:       $R \leftarrow 0, M_e \leftarrow [], t_e \leftarrow 0$
15:     **end if**
16: **until** $t > T_a$

recently generated ones. Thus, we set an expandable global memory size increasing while learning, but a fixed shared memory size with a first-in-first-out rule. Whenever new experiences arrive, the earliest saved experiences in local memory are overridden. We aim to use the experience saved in the local memory to simulate on-policy data. However, instead of exploiting a brute-force storage that indiscriminately saves every new episode experience, we set an intuitive rule to determine whether to store an experience in local memory or not. Specifically, we only save a newly generated episode from a local agent when (1) the local agent successfully accepts a policy from the global agent [2], and (2) when its episode return is not worse than the minimum of all agents' current performance. By doing so, we can avoid out-of-distribution data being saved in local memory to some extent, so as to reduce variance and stabilize learning [14]. We then allow global agents to replay experiences from the two memories drawn from a Bernoulli distribution, that is, sample experiences from the local memory with a probability $p$, and from the global memory with a probability $1 - p$. Such a Local-Global Memory Relay mechanism plays a very important role in guaranteeing global agents to consistently benefit from on-policy data as, if only a single global memory buffer is used, the probability of sampling a newly generated experience in it becomes lower and lower with more and more experiences saved alongside learning.

**Distinctive Update (DU)**: Although global agents guide local agents for exploration, each agent still maintains its own policy updating schemes to preserve learning advantages. When an agent accepts a policy from its upper-level agent, it keeps updating using its update algorithms, e.g., policy gradient, starting from the accepted policy. This is naturally achieved by the hierarchical framework stated above as well as by the performance gap determining when to transfer.

To understand CHDRL better, we provide a CHDRL instantiation, which employs a state-of-the-art off-policy agent SAC, an on-policy agent PPO and EAs agent CEM, called Cooperative SAC-PPO-CEM (CSPC). The pseudo code of the instantiated CSPC is presented in detail in algorithms 1 to 3. $G_s, L_p, L_c$ represent global off-policy agent SAC, local on-policy agent PPO and EA agent CEM respectively. Algorithm 1 shows the general learning flow of CSPC. Firstly, global agent $G_s$ is

**Algorithm 3** UPDATE
---
**Require:**
    Agent $a_\phi$, update steps $t_u$, step $t = 0$, sample probability $p$; Global shared memory $M_g$, local memory $M_l$;
1: **if** $a$ is $G$ **then**
2:     **while** $t < t_u$ **do**
3:         $o \leftarrow Bernoulli(k, p)$ with $k \in \{0, 1\}$
4:         **if** $o = 1$ **then**
5:             Randomly sample a batch $B$ from $M_l$
6:         **else**
7:             Randomly sample a batch $B$ from $M_g$
8:         **end if**
9:         Update agent's policy $\phi_s$ and value function $\psi_s$ following [25]
10:        $t \leftarrow t + 1$
11:     **end while**
12: **end if**
13: **if** $a$ is $L_p$ **then**
14:     Update agent's policy $\phi_p$ and value function $\psi_p$ following [8].
15: **end if**
16: **if** $a$ is $L_c$ **then**
17:     Update agent's new mean $\pi_{\mu_c}$ and covariance matrix $\sum_c$ following [27].
18:     Draw the current population $L_c$ from $\mathcal{N}(\pi_{\mu_c}, \Sigma_c)$,
19: **end if**
---

trained for specific steps $T_g$. This is to ensure the off-policy agent reaches a relatively good solution. Afterwards, we orderly train $G_s$, $L_p$, and $L_c$ to search the solution space for one iteration step $T$. Note that $G_s$ keeps learning from the experiences when other agents explore. After that, we evaluate the updated agent to get its new policy score $S_s$, $S_p$ and $S_c$. We then transfer policies based on these updated scores following the above principle of policy transfer. Specifically, if the score of $S_s$ is better than those of $S_p$ and $S_c$ with at least $f$ improvement, we re-initialize $L_p$ and one individual of $L_c$ with $G_s$'s policy. A similar transfer is done from $L_p$ to $L_c$.

Algorithm 1 shows what, how, and when to transfer policies, which are the three key factors in **CE**. Lines 9-13 in Algorithm 2 show how generated experiences are stored in global memory or local memory. Lines 3-8 in Algorithm 3 show how global agents replay experiences from the global and local memories. These lines combined consist of the implementation of **LGM**. Lastly, lines 9, 14, and 17 reflect **DU**, where each agent updates following its own update rules. The above procedure proceeds iteratively until termination.

Note that CHDRL also accepts the same type of agents. In this case, cooperation only exists between the global agent and local agent, not across local ones. In the ablation study, we test a case where three off-policy agents are used in CHDRL. Moreover, our CHDRL is loosely coupled in the sense that it is flexible enough to involve any other agents, e.g., DQN [19] and TRPO [24] etc., into it.

## 5 Experiments

We conducted an empirical evaluation to verify the performance superiority of CSPC to other baselines, and ablation studies to show the effectiveness of each mechanism used in CHDRL.

### 5.1 Experiment Setup

All the evaluations were done on a continuous control benchmark: Mujoco [30]. We used state-of-the-art SAC, PPO and CEM to represent the off-policy agent, on-policy agent, and EA, respectively. Note that other off-policy (e.g., TD3), on-policy (e.g., TRPO) and gradient-free agents (e.g. CEM-ES), are applicable to our framework. For SAC, PPO and CEM, we used the code from OpenAISpinningUp for the first two, and code from CEM-RL for CEM [3]. For hyper-parameters in these methods, we followed the defaults specified by the authors. For CSPC, we set the gap $f$ as 100, global agent initial learning steps $T_g$ as $5e4$, iteration time steps $T$ as $1e4$, global memory size $M_g$ as $1e6$, local memory size $M_l$ as $2e4$, and sample probability from local memory $p$ as $0.3$.

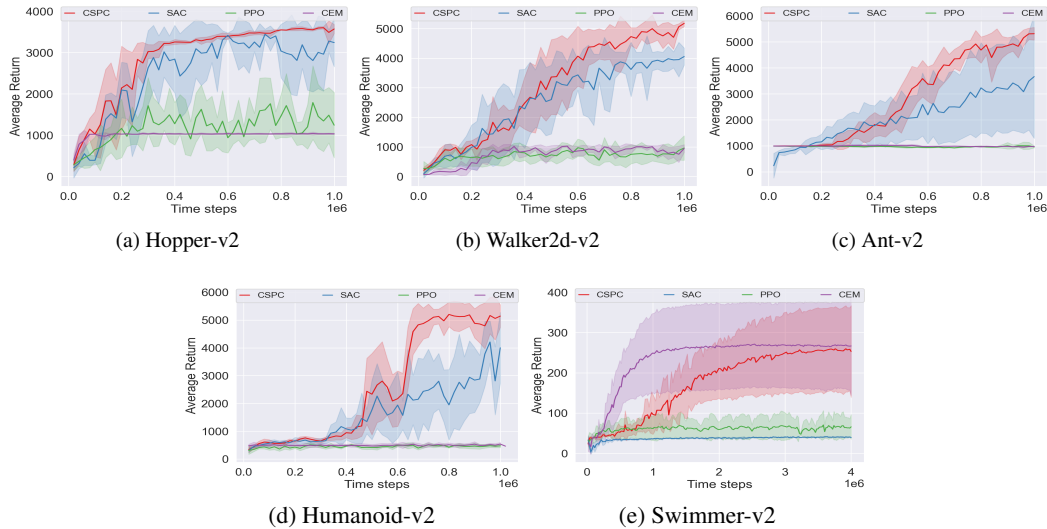

Figure 2: Training curves on Mujoco continuous control tasks.

Table 1: The max average return.

| Task | CSPC | PPO | SAC | CEM |
|---|---|---|---|---|
| Humanoid-v2 | **5412**±239 | 626±23 | 5142±133 | 616±88 |
| Ant-v2 | **5337**±220 | 1169±207 | 3766±2359 | 1019±33 |
| Walker2d-v2 | **5317**±256 | 1389±387 | 4222±290 | 1041±65 |
| Hopper-v2 | **3619**±52 | 2923±88 | 3558 ±139 | 1057±53 |
| Swimmer-v2 | 261±117 | 68±31 | 44±3 | **274**±118 |

Table 2: The elite agent.

| Task | Humanoid-v2 | Ant-v2 | Walker2d-v2 | Hopper-v2 | Swimmer-v2 |
|---|---|---|---|---|---|
| seed 0 | SAC | SAC | PPO | CEM | CEM |
| seed 1 | CEM | SAC | CEM | CEM | CEM |
| seed 2 | PPO | CEM | CEM | CEM | CEM |
| seed 3 | PPO | CEM | CEM | SAC | CEM |
| seed 4 | SAC | PPO | CEM | CEM | CEM |

## 5.2 Comparative Evaluation

We evaluated CSPC on five continuous control tasks from Mujoco in comparison to three baselines: SAC, PPO, and CEM. We also used SAC, CEM, and PPO as our candidate agents in CSPC. We ran the training process for all the methods over one million time steps on four tasks with five different seeds, and for the Swimmer-v2 task, we ran it for four million time steps. Time steps are accumulated interaction steps with the environment. For a fair comparison, we used the accumulated time steps of three algorithms used in CSPC. Specifically, we summed up each agent's time steps so that the total time-steps stayed consistent with the other baselines. The final performance was reported as the max average return of 5 independent trials for each seed. We reported the scores of all the methods compared against the number of time steps.

Figure 2 shows the comparison results for all methods on five Mujoco learning tasks. From the results, we first observe that there is no clear winner among the existing state-of-the-art baselines SAC, PPO, and CEM in terms of stability and sample efficiency. No one consistently outperforms the others on the five learning tasks. Specifically, it can be seen that, for four of five tasks (except for Swimmer task), SAC yields better results than PPO and CEM, which verifies its sample efficiency for a long run. However, we can also observe a significant variance of SAC, which indicates its high instability, especially in Ant task. In contrast, PPO and CEM have a lower variance but achieve unsatisfactory average returns. A special case is Swimmer task where both SAC and PPO fail to learn a good policy but CEM succeeds. Figure 2 also demonstrates that our proposed CSPC performs consistently better or with comparable results to the best baseline methods on all tasks. This verifies the capability of CSPC to improve the performance of each individual agent by utilizing the cooperation among them. On Swimmer task where both gradient-based methods fail, CSPC still achieves a comparable result with CEM. This is because CSPC does not benefit from SAC and PPO, and only maintains the capacity of CEM. Table 1 shows the maximum average return for each method.

One may wonder about the possible computation cost of CPSC. In our experiments, it mainly comes from the global agent, as it keeps learning for other agents' experiences in the background. The local agents run much faster than global agents, especially the CEM agent, as it is gradient-free. The total running time of CSPC is only slightly longer than the SAC agent.

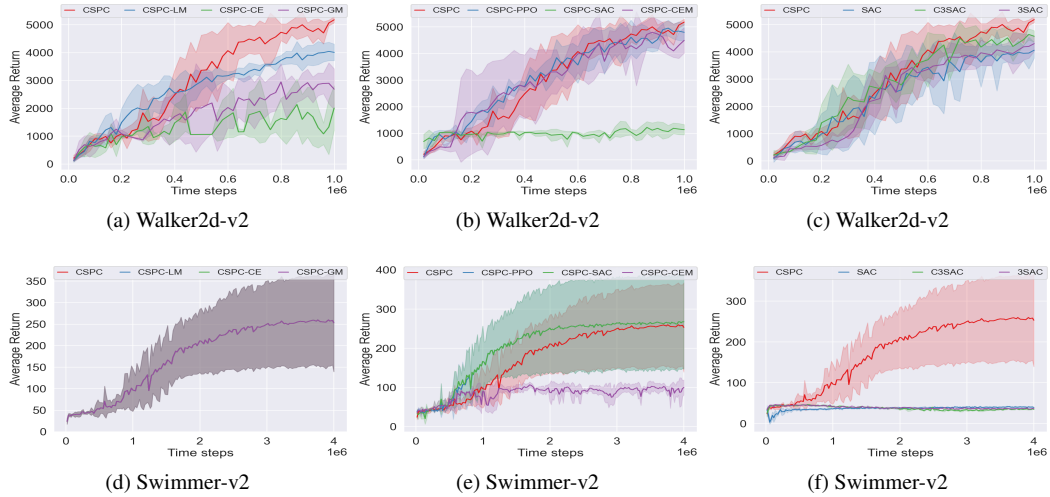

Figure 3: Ablation study on two tasks: Walker2d and Swimmer.

## 5.3 Local Agent vs Global Agent

The main motivation of this study is to figure out whether local agents really help in finding the best final policy in different random settings. To do so, we show the elite agent, that is, the agent yielding the best performance among heterogenous agents after training has terminated, in different random seeds. The results are shown in Table 2. It can be seen that CSPC could obtain different elite agents on the same task under different random seeds. Such an observation indicates that local search agents do help to find a better policy around the global guided agent. Surprisingly, the EA-based CEM agent performs better than other local agent (PPO) in most cases. However, on the complex task, Humanoid-v2, the gradient-based agents perform much better than CEM.

## 5.4 Ablation Studies

In this section, we conducted ablation studies to understand the contributions of each key component of CSPC. To do this, we built three variants of CSPC: CSPC without cooperative exploration (CE), i.e., CSPC-CE, CSPC without local memory(LM), i.e., CSPC-LM, and CSPC without global memory (GM), i.e., CSPC-GM. Specifically, in CSPC-CE, we stopped the policy transfer and let each agent explore and exploit by itself. In CSPC-LM, the off-policy agent SAC replays from all experiences uniformly. In CSPC-GM, the off-policy agent SAC only learns from its own experiences. We further analyzed the influence of each individual agent to CSPC. To do so, we developed a CSPC without PPO, called CSPC-PPO, a CSPC without CEM, called CSPC-CEM, and a CSPC without SAC, called CSPC-SAC. As CHDRL also allows the same types of agents, to verify that heterogeneous agents indeed matters, a variant of CHDRL consisting of only one type of agent was proposed. In this case, we introduced two variants: three SAC agents with CE and LGM, and three SAC agents without them. We called the former C3SAC and the latter 3SAC. For 3SAC, the three agents only shared global memory and no policy transfer existed. We evaluated all the variants on Walker2d-v2 and Swimmer-v2. The results are shown in Figure 3.

As shown in Figure 3, for task Walker2d-v2, CSPC achieved the best among all the ablation variants in terms of final average performance. From Figure 3(a), it is easy to deduce that LGM and CE indeed matter in CSPC, as without these two elements, the final performance drops quickly. From Figure 3(b), we can see that the results of CSPC-PPO and CSPC-CEM are satisfactory and only slightly worse than that of CSPC, while the result of CSPC-SAC dramatically decreases. This implies that the global agent has a more significant impact on the final performance than local agents. This is reasonable as the global agent determines the starting position of CSPC, and highly affects the following search efficiency. Note that CSPC-PPO and CSPC-CEM are CSPC without one specific local agent, but still follow CHDRL's core mechanism: CE and LGM. From the fact that their performances are much higher than CSPC-CE and CSPC-LM/GM, we again verify the significance

of LGM and CE. From Figure 3(c), we can see that C3SAC performs better than 3SAC and SAC. Even though the three agents are with the same type, local agents still provide a diverse local search as they explore in different random settings. However, our CSPC performs much better than C3SAC, while 3SAC performs only slightly better than SAC. With this, we deduce that CHDRL still improves the performance when using the same type agents, but using heterogeneous agents would further boost the performance.

For the Swimmer-v2 task, the results are different as SAC and PPO agents typically fail on this task. In other words, the global agent is incapable of finding a relatively good position, and only the CEM agent works. The most likely explanation is that in Swimmer-v2, existing DRL methods provide deceptive gradient information that is detrimental to convergence towards efficient policy parameters [21]. Hence, LM/GM/CL cannot enhance the final performance, which is shown by CSPC-LM,CSPC-GM and CSPC-CL in Figure 3 (d). In such a case, the learning curves of the three methods mostly overlap. On the other hand, CSPC-PPO and CSPC-SAC gain a better final performance than CSPC, which is also reasonable as the CEM agent has more iterations leading to a better final performance, as shown in Figure 3(e). For the same reason, C3SAC and 3SAC both fail.

# 6   Conclusion

In this paper, we present CHDRL, a framework that incorporates the benefits of off-policy agents, policy gradient on-policy agents and EAs agents. The proposed CHDRL is based on three key mechanisms, i.e., cooperation exploration, local-global memory and distinctive update. We also provide a practical algorithm CSPC by using SAC, PPO, and CEM. Experiments in a range of continuous control tasks show that CSPC achieves a better or comparable performance compared with baselines. We also note that CHDRL introduces some new hyper-parameters which may have a crucial impact on performance, however, we do not tune that too much. Moreover, we should carefully select the agents, as the final performance highly depends on the agents used, particularly the global one.

## Broader Impact

The DRL agent that learns from an incompletely known environment runs the risk of making wrong decisions. This could lead to catastrophic consequences in practice, such as automated driving, the stock market, or medical robots. One approach to alleviate this risk is to combine with other techniques or involve human beings' supervision. In terms of benefits, DRL can be deployed in a safe environment where a wrong decision will not lead to a significant loss, e.g., the recommendation system. Moreover, in some environments that we can simulate well, it would be very promising to develop an intelligent robot to work in such an environment.

## Acknowledgments and Disclosure of Funding

This research is partially funded by the Australian Government through the Australian Research Council (ARC) under grant LP180100654.

## Footnotes

[1]Policy transfer only happens from upper-level agents to lower-level agents.

[2]It ensures the local experiences are close to the on-policy data of the global agent's current policy.

[3]OpenAISpinningUp: github.com/openai/spinningup; CEM-RL:github.com/apourchot/CEM-RL

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
