[Reviews · NeurIPS 2020]

Review 1

Summary and Contributions: --- UPDATE: The authors resolved / explained most of my concerns, some of which were caused by my not-complete undertanding of the presented concepts. The experiments seem to be reported fairly, wrt. the time-steps used. There seems to be no prior work on ensemble of heterogenous RL-agents, meaning this work has a potential to inspire a new direction of research. I still think that the paper would benefit from proof-reading and some more "polishing". I suppose the author will revise the text itself with some of the explanations. With that in mind, I feel that the paper is of interest to the community, sufficiently novel and can be accepted (reflected in the score change). --- The paper presents a framework to combine several distinct RL algorithms together for better learning. A single "global" off-policy agent learn from an experience buffer which is updated with other "local" on-policy agents, which are periodically updated to match the global agent's policy.

Strengths: The idea of combination of several algorithms together to extract their strengths is interesting.

Weaknesses: The paper lack some important details. The exact mechanic of the policy transfer between different algorithm is not given. Given the content, I may assume that "transfer" means a simple copying of the parameters, but I remain unsure. When augmenting the experience buffer with other algorithm, it would be nice to clarify why it does (not) introduce any bias in the data. It seems that the different parts of the framework could be replaced by a different way of "tinkering" with a algorithm or its hyperparameters. E.g., the auxiliary on-policy algorithms are here mainly for exploration, but the exploration of the main off-policy algorithm itself can be easily controlled and I suspect it can, with the right setting, work as good as the given complicated framework. The global and local experience buffer seems more like a hack. Again, with the original algorithm, the size of the experience replay buffer can be easily controlled. These settings are not explored, and it is unclear whether they can or cannot substitute the given framework.

Correctness: The experiment setup does not describe the exact way of reporting the results. The X-axis on the graphs is labeled "time-steps", which I am unsure what it means in this context. The proposed framework consists of 3 algorithms trained together, which generate 3-times more "time-steps", which I am unsure if they are accounted for. The source code is not given to verify.

Clarity: The paper would benefit greatly from proof-reading, or at least using some typo-checking service. It contains a lot of errors, which impede the understanding of the content. The method of tranfering the policies is not explained at all. There's a section called "How to Transfer" which do not give any details on how to transfer. In Preliminaries, the authors give a summary of the common algorithm too concisely. More explanation would help the reader. The first sentence of the abstract is utterly superfluous. Overall, the paper need some more work on its style.

Relation to Prior Work: There is a general related work of RL. No work combining different algorithms is presented (I do not know any, but suspect that there may be. If not, state it clearly).

Reproducibility: No

Additional Feedback: "Cooperative" in the title suggest a multi-agent RL setting, which may confuse the reader. Consider a different name. It would be great to include some description in the figures' captions. E.g. Figure 3d looks weird, and the description why is given in a block of text, hardly to find. I am unsure how the "Broader Impact" section discussion is related to the presented framework.


Review 2

Summary and Contributions: This paper proposed the CHDRL framework to achieve high sample efficiency and maintain good stability by cooperative learning of a group of heterogeneous agents (off-policy, on-policy, and population-based EAs). The three main mechanisms behind CHDRL, i.e., cooperative exploration (CE), local-global memory relay (LGM) and distinctive update (DU) are well illustrated and supported by an instantiation (CSPC) using a group of state-of-the-art off-policy, on-policy, and population-based EAs agents. Experiment results on a range of continuous control tasks show that CSPC achieves higher performance than each state-of-the-art agent from the aforementioned group, respectively. An ablation study on three main mechanisms of CHDRL is also provided. =========================== Update after rebuttal: I still have concerns about the practical usage of the proposed framework in a wider range of tasks, mainly on the choice of extra hyperparameters. I would like to keep my original score.

Strengths: 1) Combining the advantage of off-policy, on-policy, and population-based EAs RL algorithms is an interesting and important topic. 2) The CHDRL framework is technically sound and well-illustrated. The proposed framework appears to be novel.

Weaknesses: 1) The performance of the instantiation of CHDRL framework, i.e., CSPC, heavily relies on the choice of hyperparameters, including that of each heterogeneous agent. However, the paper did not provide enough information on how to choose them. This could limit the practical usage of CHDRL. 2) It would be better to include theoretical analysis, e.g, the convergence guarantee, since the main contribution is a general framework.

Correctness: No explicit incorrect statements.

Clarity: The paper is written clearly.

Relation to Prior Work: Yes.

Reproducibility: Yes

Additional Feedback:


Review 3

Summary and Contributions: This paper presents a Cooperative Heterogeneous Deep RL(CHDRL) framework to learn a policy by incorporating the benefits from different agents with off-policy, on-policy and evolutionary policy learning. The global agent uses off-policy learning and transfers the knowledge to local agents. While local agents adopting on-policy or evolutionary policy learning to explore the local area. Experiments on MujuCo environment demonstrates the CHDRL is able to achieve the better performance.

Strengths: 1. Cooperative Learning It is interesting to introduce the framework of cooperative learning by incorporating benefits from global and local agents with different policy learning methods to transfer knowledge and trade-off exploitation and exploration. 2. Local-Global Memory Reply The global memory is served to the exploration experience for all agents, while local memory for the simulation of on-policy data. This memory replay is important for the global agent to keep on learning efficiently.

Weaknesses: Possible Computation Cost As we need to learn the policy with more than one agent based on different off-policy, on-policy and evolutionary strategies, the computation cost might be larger than a single agent.

Correctness: The CHDRL algorithm is the combination of different existing policy learning methods. It is an empirical algorithm without any theoretical proof. However, the experiments show that the algorithm is able to achieve reasonable good empirical results.

Clarity: Yes, the paper is clearly and well written. It is not difficult to understand and follow it.

Relation to Prior Work: Yes. Compared to guided policy search or evolutionary RL, CHDRL can benefit not just from off-policy and evolutionary learning, but also on-policy learning schemes. Unlike A3C and IMPALA, CHDRL has global and local agents, and mainly focuses on the cooperation of diverse learning schemes.

Reproducibility: No

Additional Feedback: I have read all the reviews and rebuttal, the authors have addressed my concern. Yes, I keep the score 6.


Review 4

Summary and Contributions: The paper presents a new deep RL framework that combines on-policy off-policy and Evolutionary Algorithms by using 3 key mechanisms, i.e., cooperation exploration, local-global memory and distinctive update. As a sample algorithm, the authors combine SAC, PPO and CEM. The authors evaluate their method on 2 Mujoco tasks and conduct ablation studies. The results are comparable with other methods. In my opininon, the strongest result of the method is that, it performs good on all tasks. While different algorithms fail at different tasks, the provided algorithm benefits from using all 3 methods and converges to good policies in all tested problems.

Strengths: In my opinion the strongest part of the paper is the ability to combine 3 types of algorithms, and the framework is method-agnostic. Although the authors combine SAC, PPO and CEM, it's possible to replace them with others. Another strength is finding solutions in all the problems where some of the algorithms fail to converge on particular problems. In a sense, the framework benefits from all 3 types of learning, in one type does not work, it can still learn.

Weaknesses: The results on mujoco tasks are not that impressive compared to state of the art. They are equivalent or pretty close to other algorithms' performances. IN addition, the proposed framework introduces many new hyperparameters. Although these hyperparameters can provide better results, they require further optimization by the user. Sometimes simpler algorithms are more preferrable in this context.

Correctness: The method and the methodology seems to be correct. I couldn't understand if the figure 3d is missing data or not.

Clarity: Paper is clearly and well written. The authors give detailed intro to different RL algorithms and latest research related to their method. The algorithm is explain well as well.

Relation to Prior Work: The paper clearly discusses the difference of this work compared to previous works. They also provide detailed evaluation on comparison of these methods and ablation study.

Reproducibility: Yes

Additional Feedback: -

[Author Response · NeurIPS 2020]

We thank the reviewers' valuable comments. Our responses are listed below.

**To Reviewer 1:** (Q1)"...mechanic of policy transfer is not given,..., transfer means a simple copying of parameters...".

**A**: We want to highlight that CHDRL follows the basic mechanism of transfer learning [ref1] including 3 essential points, i.e., *what, how, and when to transfer*. We handle all the 3 points in policy transfer among heterogeneous agents. The difficulties and the corresponding solutions are elaborated in Lines 136-160. "simple copying of the parameters" is only a small portion of *what to transfer*, and actually CHDRL does more than copying of the parameters as different agents have different policy structures which makes the simple copy infeasible.

(Q2)"augmenting the experience buffer with other algorithm, ...,clarify why it does (not) introduce any bias in the data."

**A**: Global agents are not simply replaying experiences from the global memory that augments experiences of different agents. Instead, they replay experiences from global and local memory buffers following a probability distribution, which alleviates the bias caused by replaying global memory only. Moreover, for local memory, intuitive rules are set to store local agents' experiences as elaborated in Lines 173-175. The rules guarantee that local memory only saves experiences similar to the global agent's current policy, which further reduces bias, as evidenced in [14].

(Q3)"... be replaced by a different way of' tinkering 'with a algorithm or its hyperparameters,... on-policy algorithms are here mainly for exploration, but..., with the right setting, work as good as the given complicated framework.". **A**: As evidenced by [6,14], off-policy agents suffer from bootstrapping error and extrapolation error regardless of how hyper-parameters are fine-tuned. It is also well known that Off-policy Q learning is not to converge even with linear function approximation [ref2]. Thus, to the best of our knowledge, it is extremely hard to overcome these issues by "tinkering." These points are our motivations to integrate different methods. Moreover, except for benefiting exploration from on-policy agents, we also maintain other on-policy agents' advantages, e.g., stability, in CHDRL, as shown in Lines 65-74. Verified by Section 5.3, on-policy agents do help find a better policy around the off-policy agent.

(Q4)"...the size of the experience replay buffer can be easily controlled. These settings are not explored, ...".

**A**: The intuition here is to enable off-policy agents to benefit from diverse local experiences so that it can make more progress [6,14]. Uniformly replaying experiences from global memory that augments all the experiences of different agents definitely fail to do so. Thus, we propose a global-local memory buffer and employ a rule to save the experiences that are similar to the off-policy agents' current policy in local memory, see Lines 172-175. The size of the local memory is fixed. We don't specifically fine-tune it in the experiment, and already achieve good improvements. The global memory is similar to the conventional RL methods' setting, i.e., SAC, and we simply follow the default configuration.

(Q5) "..,X-axis on the graphs is labeled 'time-steps'...,generate 3-times more "time-steps", ... source code is not given."

**A**: Time-steps are accumulated interaction steps with the environment. For a fair comparison, we use the accumulated time steps of 3 algorithms. Specifically, we sum up each agent's time steps so that the total time-steps stay consistent with the other baselines. We will publish our source code.

(Q6) "...'How to Transfer' which do not give any details on how to transfer." **A**: We elaborate on the policy transfer from *what, how, and when to transfer* in section 4. How to transfer focuses on the transfer methodology we employed, i.e., a hierarchical transfer manner: 1) off-policy agents to both on-policy agents and EAs agents, and 2) from on-policy agents to EAs agents. We guess the reviewer may question on *what to transfer*. We transfer $\mu_\phi(s)$ among different agents. Line 10-19 in Alg 1 clearly shows *what, how, and when to transfer*.

(Q7)"...No work combining different algorithms is presented...". **A**: To the best of our knowledge, this is the first work combining heterogeneous agents, including off-policy, on-policy, and Evolutionary-based RL methods.

**To Reviewer 2:** (Q1) "...choice of hyperparameters... the paper did not provide information on how to choose..."

**A**: For CHDRL, we did not specifically fine-tune the hyperparameters and already achieved good results. We believe the performance can be further boosted by fine-tuning the hyper-parameters.

(Q2)"...better to include theoretical analysis, e.g, the convergence guarantee,...".

**A**: We agree that theoretical analysis makes the work more solid. We will work on this in future work.

**To Reviewer 3:** (Q1)"Possible Computation Cost...". **A**: The computation cost of CHDRL mainly comes from the global agent, which is comparable with SAC in our case, as it keeps learning for other agents' experiences in the background. The local agents run much faster than global agents, especially the EA agent, as it is gradient-free.

**To Reviewer 4** (Q1)"The results on mujoco tasks are not that impressive compared to state of the art..., these hyperparameters can provide better results, they require further optimization by the user...".

**A**: As shown in Figure 2, CHDRL achieves clear improvements on the first four tasks. Regarding the mean of the max average return for the four tasks, CHDRL is 4921.25, which outperforms PPO (1526.75) 3394.5, SAC (4172) 749.5, and CEM (933) 3988.25. For the 5th task where conventional RL methods fail, CHDRL still achieves comparable results with CEM, as SAC and PPO do not help, CHDRL only reflects the advantage of CEM. Moreover, we did not fine-tune the hyper-parameters. We believe CHDRL can achieve better results after carefully fine-tuning.

(Q2)"I couldn't understand if the figure 3d is missing data or not.". **A**: It is because LM/GM/CL cannot enhance the final performance in task Swimmer, and the learning curves of the three methods are mostly overlapped. In this case, the gradient-based agents fail to learn, and only gradient-free CEM agent works.

[ref1] A Survey on Transfer Learning.

[ref2] Residual Algorithms: Reinforcement Learning with Function Approximation.

[Meta-Review · NeurIPS 2020]

Following the rebuttals, all four reviewers agreed that this paper should be accepted. While there are remaining questions around the hyperparameters (and performance relative to other methods), and computational cost, this is an interesting and novel line of work. The authors are encouraged to proofread the paper thoroughly and address the issues raised by the reviewers.